# Rethinking Sense of Place Interpretations in Declining Neighborhoods: The Case of Ami-dong Tombstone Cultural Village, Busan, South Korea

**Sreenidhi Konduri** * and **In-Hee Lee** *

Department of Architecture, Pusan National University, Busan 46241, Republic of Korea
* Correspondence: nidhi23@pusan.ac.kr (S.K.); samlih@pusan.ac.kr (I-H.L.)

**Abstract:** In recent years, urban regeneration strategic plans have been implemented across South Korea to curb the negative impacts of depopulation, physical deterioration and economic decline. By adopting a people-centered regeneration process, context-sensitive plans are formulated by integrating local people's perceptions and expectations. This paper examines urban regeneration plans implemented in Ami-dong Tombstone Cultural Village, a declining hillside village in Busan, using "people–place–process framework of place attachment" to analyze the process of articulating "sense of place" through community-driven approaches. Based on archival research, site investigations, secondary data and semi-structured interviews, the paper explores the revival of social capital, integration of place-sensitivity and improvement of everyday landscapes through affective, behavioral and cognitive responses of urban professionals and community members involved in the project. Findings of the study show that place attachment, sense of community and community participation in regeneration can help in re-creating meaningful places. Lessons from Ami-dong offer insights on ways to strengthen people–people and people–place relationships through design and planning processes in a declining context with an aging population.

**Keywords:** sense of place; urban regeneration; people–place–process framework; community engagement; Ami-dong





## 1. Introduction

The importance of sense of place in design and planning has long been researched and proven to be a principal consideration for designing meaningful environments [1,2]. Place sensitivity is "something beyond the physical and sensory property of a place" [3] and can be translated from the character of the place and its people, their whereabouts, daily experiences, relationships and feelings bound to the place [4,5]. In essence, the experiential context in a physical setting generates a "sense of life" [6], realized through people's actions [7], and is integral to our existence in the world [8]. Recent studies outline the people–place connections transcending beyond physical determinism by exploring a multiplicity of social, cultural and psychological place dimensions [9]. The quality of making sense of place, or what Carmona refers to as "placefulness" [3], has been explored through functional concern or place dependence [10], rootedness or place attachment [11], nostalgia or place memory [12], physical/social association or place identity [13], quality of place or place value [14] and social capital or sense of community [15].

While these place-related constructs have been widely explored, the translation of people's perceptions of a place into practice has been limited due to numerous definitions and conceptualizations [16] and overall complexity of the process [17]. The increasing intricacy of this issue is a consequence of focusing exclusively on why people's perceptions are important in designing environments or how does place-making impact people's experience, cognitions, emotions and behaviors. However, the main component missing from this myriad of discussions is finding concrete cases that narrate the process of integrating

place sensitivity in designing good places. Particularly, the contemplation of people–place relations within the context of disruptions and urban changes remains underrated by a common misconception that individuals living in such unsettling situations often have lower rates of place attachment [18]. Despite these claims, people continue to remain attached to their environments in the face of demographic decline [19], urban change [18], migration [20], relocation [21], disaster [22] and conflict [23].

To address these gaps, this article illustrates an empirical study of sense of place elements by adopting the "people–place–process framework of place attachment" proposed by Scannell and Gifford [24] and provides a constructive dialogue of "making sense of places" to "(re)make places". Considering the challenges of urban decline and the aging population in Korea, the article contemplates sense of place as a prerequisite for creating an authentic physical setting against the backdrop of the declining and changing landscape of Ami-dong Tombstone Village, Busan. Unfolding the process of place-making in the context of disruptions ranging from the historic refugee crisis to the recent depopulation and decline, the article will reveal how people live(d) alongside urban stresses and continue to lead a harmonious life by creating their own place within this chaos. The findings offer an understanding of integrating people's aspirations, experiences, memories, values and assets within different layers of the urban fabric in urban regeneration projects implemented over the past two decades.

### 1.1. Why Should we Make Sense of Places?

In "The Timeless Way of Building", Christopher Alexander describes an unnamed quality in every person's narrative found in the liveliest of moments [7]. Relph illustrates this ambiguous quality as insideness, which transforms a physical setting into a place [25], inducing a feeling of "being-at-home" [26]. The very innateness of this quality is perceptually taken for granted [27] as it is expressed in the "affirmations" of everyday lives, revealed through awareness of space, mood, character and behavior [28]. These "situated life episodes" reveal how people make sense of place as they are deeply attached, involved or concerned about their physical context [27,29]. Thus, "sense of place" is a human response to the environment communicated through everyday life and experiences [30]. It is a dynamic human–environment relationship involving emotions, feelings and experiences within a place [31]. The consensus of these interpretations can be found in the description of a good place by Kevin Lynch [6]:

> So I risk a general proposition: a good place is one which, in some way appropriate to the person and her culture, makes her aware of her community, her past, the web of life, and the universe of time and space in which those are contained.

### 1.2. How Can we Communicate "Sense of Place" Constructs through Physical Environments?

In the recent past, there has been much consensus on the effect of design processes on the local sense of place and resident's place attachment [32,33] and, conversely, on identifying place-based perceptions that can be employed in designing meaningful physical environments [17]. However, the plethora of approaches for depiction and measurement of people–place relationships and agreement on a particular construct (place identity, place attachment, sense of place, place dependence) has been long debated [12], owing to the multiplicity of definitions and approaches adopted across different disciplines. As Lewicka argues, with abounding research, can we state the actual reasons that cater to the relationship between people and places [16]?

While other disciplines have focused on quantitatively measuring people–place relationships, much of design and planning literature has focused on constructing conceptual and theoretical frameworks [1,5] or phenomenological explorations [34,35]. However, little attention has been paid to the practical application and outcomes of these frameworks. Especially within the context of disruptions and place changes, the application of these concepts has been disregarded as research shows that people living in places with incivilities and environmental disruptions have lower rates of place attachment [36]. On the contrary,

even in sensitive contexts, place association can be a focal point for improving places and vice versa [19].

One study based on community-led regeneration in China reveals that strong social ties and communal identity in a deteriorating fishing community helped to revitalize the physical fabric through participatory planning and design [37]. In Korea, collaboration of artists, residents and local government in culture-led regeneration programs helped in building neighborhood trust, revitalizing the physical fabric and strengthening people–place relationships, which encouraged the residents to actively participate in future improvement programs [38]. Another study related to the disorientation and reorientation among people after the F3 Tornado in Goderich, Ontario shows that the people–people and people–place relations were strengthened after the disaster through active and collective engagement of the community in the recovery and restoration process [22]. Studies on involuntary relocations due to conflict or development and rational planning for place improvements have shown that strong place associations have helped in re-establishing a sense of local identity despite the place transformations resulting from urban regeneration [18,21,23]. Therefore, irrespective of the contextual setting (ordered or disrupted), the relationship between people and places can be analyzed by examining the organization of space, meaning, time and communication within the perceived environment [39].

The quest for this inquiry starts by identifying environmental cues within everyday settings that explain "who does what, where, when, how and including or excluding whom" [40]. Such patterns are based on the social membership or collective identity characterized by people's daily routines and use of the environment, bonding with the place, interactions, practices and symbols within a place [15]. This membership enables a sense of community or togetherness that motivates people within a place to voluntarily participate in improving and maintaining their habitus [41,42]. Scannell and Gifford define such affiliations of an individual or group with a certain place based on the physical and social character of the place communicated through affective, cognitive and behavioral processes using the "People–Process–Place framework of place attachment" [24]. The framework integrates a diversity of psychological, social and physical conceptualizations and is relevant to applied settings that rely on the strength of people–people and people–place relations.

By expanding the functions within this framework using various environmental cues suggested in previous research, this article will analyze the importance of sense of place and place perceptions in design and planning processes in community-centered urban plans (see Figure 1). This discussion will consider three future directions of people, place and process dimensions, suggested by Lewicka in her review of 40 years of literature on place attachment as seen in Figure 1 [16]: (1) people—the role of "social capital" in enhancing place-related emotions and community empowerment for protecting and maintaining neighborhoods; (2) place—the physical expression of values, meanings and experiences of the people through the "built environment"; and (3) process—to recognize the processes through which people form emotional bonds with places and reinterpret them through a multidisciplinary framework [16].

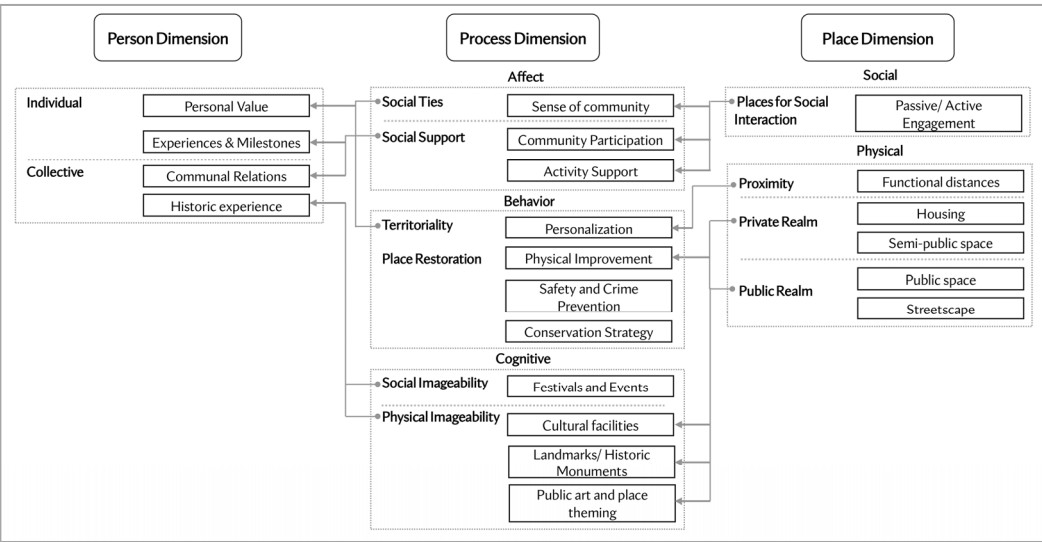

**Figure 1.** People–Place–Process Framework of Place Attachment.

## 2. Methods

This study seeks to understand how design and planning processes have helped in preserving "sense of place" through urban regeneration in the disrupted landscape of Ami-dong Tombstone Village, Busan, South Korea. First, based on archival research, the historic development of the village will be explored to understand the physical and socio-cultural context during different periods. Second, to obtain information regarding the design and planning process, semi-structured interviews of the project chief architect, public officials and local community group leaders and members were conducted. Based on these discussions and site investigations, the study will explore how different elements of sense of place have been contextualized, through dialogue and narration, in urban regeneration plans during the recent decades. Lastly, the study will conclude by explaining how architects/urban professionals have integrated place perceptions within urban regeneration plans, the implementation process and outcomes of the place-sensitive design/planning.

## 3. Case Study

### 3.1. Study Area

Many cities across Korea have shifted focus from development to regeneration, responding to the challenges of urban decline. In 2013, the national government implemented the "Special Act on Promotion of and Support for Urban Regeneration" for collaborating all levels of government and local stakeholders to form an integrated network to formulate urban regeneration plans. Busan was one of the first cities to implement a "people-centered urban regeneration" approach for the regeneration of "Sanbokdoro Maeul" ("hillside villages") under the "Sanbokdoro Renaissance Project" [43]. With the successful regeneration of Gamcheon Culture Village through this project, which has gained global attention during recent years, the city and local governments continue their regeneration efforts to improve other declining neighborhoods through community, culture and art [44]. Based on the case of one of the recent project areas, Ami-dong Tombstone Village, a hillside refugee village[1] in the original city center[2] (see Figure 2), this study will explore the process of urban regeneration through community participation and culture-led regeneration strategies. Currently, the village population is 7566, out of which 30% (2213) are aged 65 years and above. Ami-dong presents a special case since the regeneration projects implemented during the past 10 years articulate residents' perceptions through place sensitive design/planning approaches considering the ongoing challenges of aging population and demographic decline.

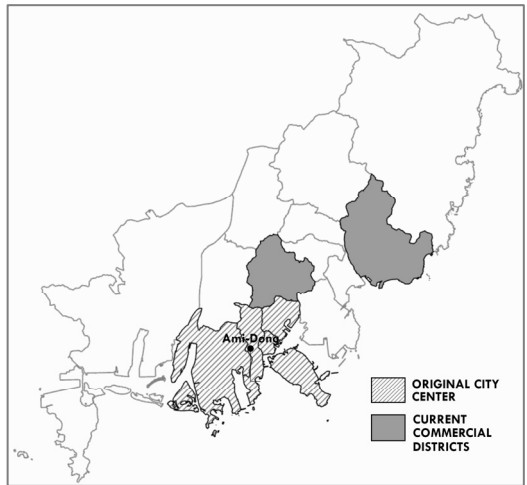
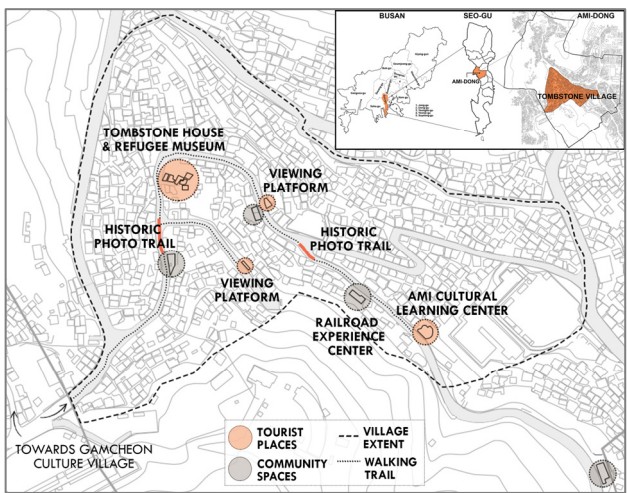

**Figure 2.** Map of Busan showing the Original City Centre (**Left**), and Location of Ami-dong and places created/re-created between 2013 and 2020 (**Right**).

*3.2. History of Ami-dong Tombstone Village*

Ami-dong has long endured physical transformations during different periods in the contemporary history of Busan. Tracing back to the Japanese Colonial period (1910–1945), Ami-dong started as a shantytown built by rural migrants and was later designated as a Japanese cemetery by the city authorities. The cemetery was abandoned after the Liberation (1945), and with the sudden outbreak of the Korean War (1950–1953), Ami-dong was designated as one of the temporary refugee settlements, where large tents were laid across the cemetery to accommodate refugee families. Under circumstances of no return after the Armistice Agreement (1953), North Korean refugees and people who lost their families during the war stayed back, transforming the cemetery from a temporary refugee camp to a permanent living space. The village population continued to grow as more evacuees from other shantytowns were relocated to Ami-dong, owing to frequent fire accidents or forced demolition of make-shift shelters by the city authorities [45].

In the post-war years, Busan became an industrial hub with the development of labor-intensive factories attracting many people from the surrounding rural areas, resulting in a threefold increase in population from 1.2 million (1960) to 3.15 million (1980) [32]. Ami-dong once again gained the attention of the migrant population, owing to the low rental prices on account of illegal and unauthorized constructions due to the lack of proper administrative setup in the village. In the 1970s, a decentralized administrative division was established, and the village was divided into two dongs (administrative unit for a neighborhood in Korea). National-level policies, including Housing Improvement Policy (1973) and Temporary Measures Act on the Promotion of Housing Improvement (1973), helped to improve the residential environment in the village. However, over 90% of the houses built in the village were found to be illegal constructions as the land remained under cemetery land use and belonged to the local government. To control the overwhelming number of illegal constructions, the local government restricted new constructions within the area and only permitted repairs or restoration. Under the "Act on Special Measures for Regularization of Specific Buildings" (1981), few unregistered properties were legalized [46]. Towards the end of the 1980s, the village growth stagnated slowly as the incoming migrants preferred to live in the adjacent Gamcheon-dong Taegukdo Village with better facilities and planned infrastructure [45].

During the early 1990s, deindustrialization, accompanied by the shift of city development towards the east and new town development, resulted in the decline of old downtown areas including Ami-dong. Even as the urban services in the village improved during the mid-1990s [46], a large proportion of houses in the village were abandoned or left vacant, increasing concerns about safety and hygiene. In the early 2000s, the negative effects of

depopulation were clear from the rising numbers of elderly population, economic distress and physical deterioration. In response, regeneration initiatives were implemented under the Sanbokdoro Renaissance Project (2010), which helped in transforming the physical and social environment in Ami-dong by renovating the existing built environment and preserving the cultural landscape through resident participation [44]. Despite these efforts, the village population declined further from 10,549 in 2010 to 9022 in 2015 [47]. In 2015, the local government proposed the Ami-Chojang Urban Regeneration Project (2015–2020) to strengthen the local community and further implement strategic plans for physical, social and economic revitalization [48]. Although the results have been lower than expected, the local government and urban professionals continue their revitalization efforts, taking note of the historical significance of the village and the residents' sense of place.

### 3.3. A Potpourri of Senses of Place—People-Centered Urban Regeneration

Based on the discourse of content in the people–place–process framework, the following section will discuss the processes engaged in promoting individual/collective level actions for place changes (regeneration) in public, semi-public and private realms of the village. The people and place dimensions will be explained in connection with the sub-dimensions of the process dimension (as seen in Figure 1).

3.3.1. Affective Responses—Social Capital and Collective Efficacy
Sense of Community, Community Participation and Activity Support

The dark history associated with war and death and the irregular topography of Ami-dong have translated into disconnected communal relations and formation of segregated residential pockets. Even within these pockets, seclusion is pronounced further by the vacant and abandoned houses cast by population decline and increase in aging population during recent decades. Consequently, in the late 2000s, the village was in desperate need of revival of social capital and community empowerment to tackle the decline. The local government established a resident-led urban regeneration initiative in 2010 that helped to bring together residents to plan and manage small-scale projects alongside design professionals and other stakeholders. These interventions mostly prioritized physical renewal, and later in 2015, the focus shifted towards community development to encourage the residents to collectively work for the betterment of their own living environment. Community-level meetings were organized to give the residents a "voice" to express their concerns and needs and a "choice" of what kind of places/activities they were trying to seek. Following the preliminary survey and capacity-building programs, an integrated village network of residents, urban professionals and other stakeholders was created for initiating the regeneration projects.

The chief architect heading the project conveyed:

By organizing community meetings on regular basis, we were able to identify the actual issues and concerns of the residents. The local leaders helped in gathering people for these meetings and soon we were able to successfully draw out proposals for regeneration interventions together with the residents.

Notably, the "Ami-Moms" community, formed by the mothers of the children studying in the village elementary school, has played an active role in strengthening the village community after noticing the closure of private education institutes and extracurricular activity centers and the lack of outdoor facilities for children in the village. In response, small-scale children's play areas were designed in the vicinity of the school. The Moms' community includes one head organizer and 10–15 members aged between 30 and 50 years who were mostly homemakers. All the members of this community were given training sessions on organizing children's activities, cooking/baking and marketing techniques for promoting local businesses. The local government provided the Railroad House Art Experience center, a café-cum-community space for the group to organize community programs and activities for children and the village elderly (see Figures 3 and 4). The community is self-sustainable with the café, a small baking business and a guesthouse.

The moms' group has continued its efforts to activate the social capital of the village by motivating and encouraging participation in activities and programs. The moms' community head, a lifelong resident, conveys the importance of village empowerment and rebuilding social capital in the regeneration process:

> We wanted to help our people and make our village a better place to live. The regeneration project really helped us in every possible way to achieve that. Today we are much more united, and our village looks a lot better than before.

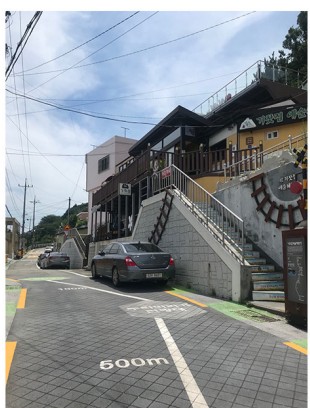
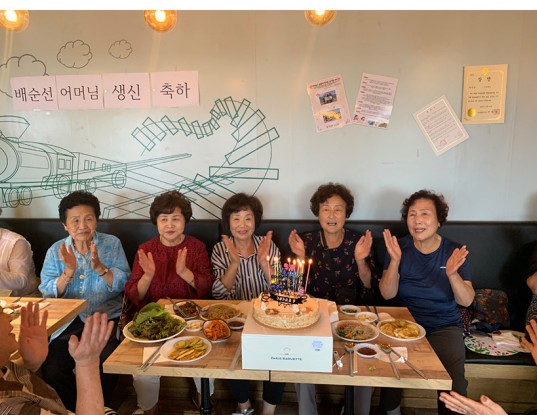
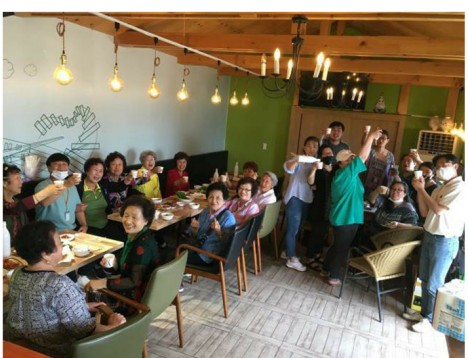
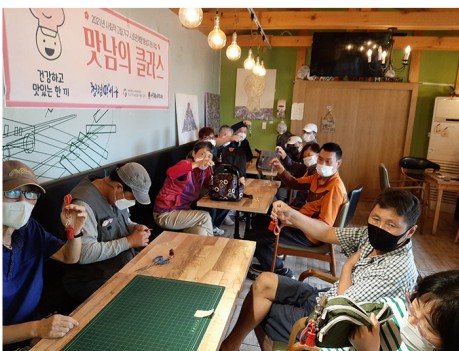

**Figure 3.** Railroad House Art Experience Center and programs organized by Ami-Moms.

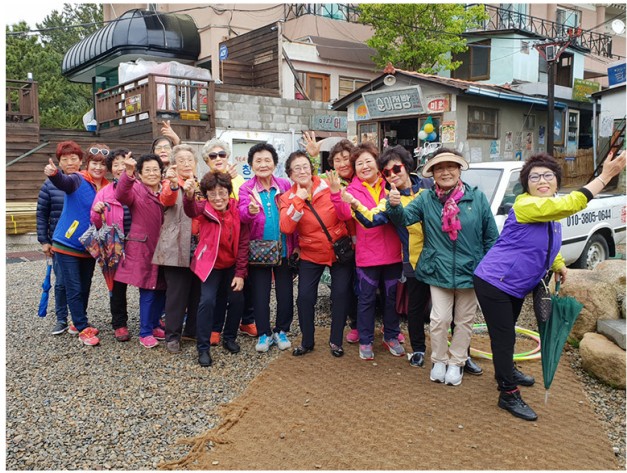
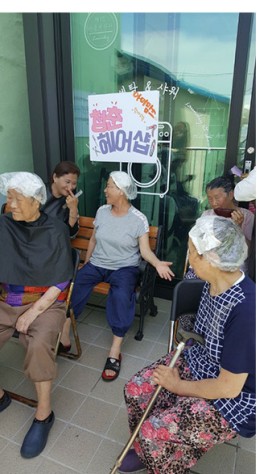
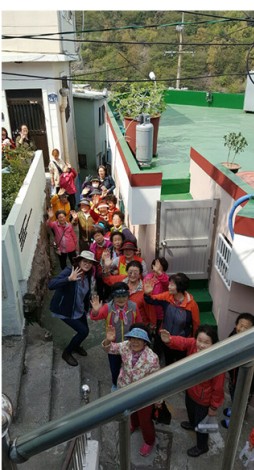

**Figure 4.** Other programs organized by Ami-Moms in the village.

The Busan city and local government have built multi-purpose community centers that provide public space and services to meet the social and recreational needs of the residents. The "Happy Village Community Centre" is one of the most successful communal facilities in the district for promoting resident-led activities and regular interaction among

the residents. Considering the physical constraints owing to the steep-sloped topography of the area, the regeneration project helped in providing several other small-scale community centers to reach every resident living in the vicinity. The creation of such social spaces within walkable distances has revived a strong sense of community, feelings of mutual trust and shared community welfare in the village.

3.3.2. Behavioral Responses—Place Restoration

Physical Improvement (Private Realm—Housing)

Due to restrictions on new constructions since the late 1980s, the housing condition deteriorated over time and has worsened with the demographic decline. Clusters of vacant and abandoned houses in every residential pocket raised concerns of safety and hygiene. Most of the dwellings lack sewer lines as they remain inaccessible due to higher slope and lack of legal status. Improvements were planned to address the issues identified by the residents during the initial survey. Plans for repair/restoration of old houses, demolition of abandoned houses, construction of community toilets and creation of semi-public spaces and shared public facilities were implemented. Practical training sessions on low-cost eco-friendly remodeling for basic housing repairs, insulation, waterproofing and other installations helped to upgrade the physical context based on the residents' willingness and encouraged residents to participate in the improvement process.

Physical Improvement and Personalization (Semi-Public and Public Realms)

In the years following the Korean War, the refugees in the village constructed houses with the tombstone as a base[3] and gradually improved the structure over the following decades (see Figure 5). The tombstone restricted the floor area in most of the houses to less than 20 m$^2$ and, in some cases, close to 30 m$^2$ [45]. The houses are too small and lack enough space to accommodate a washing machine or even a room heating system. To tackle this issue, few of the abandoned houses were demolished to make a "Ssamji-Madang" ("a semi-public urban pocket") for a small cluster of houses. This semi-public courtyard acts as an extension that allows the residents to get an additional space complementing the functions absent in the dwelling space and encourages proactive neighboring. Presently, the space serves multi-purpose functions (drying laundry, growing small plants, social gatherings and as a common dining area). The residents are responsible for maintaining this space and its surroundings. The collective ownership of this shared space establishes a sense of togetherness as residents perceive it as "our" space rather than "my" space. During the resident survey, close ties were noticeable, especially among the elderly women in the village, who often meet each other, talk and eat together. These social relations remain invisible to visitors as they remain restricted to small pockets of neighboring. These semi-public urban spaces designed during recent years have encouraged routine interaction, further strengthening the existing relationships. Other facilities, such as a laundry house and a hair salon, were also provided by remodeling vacant houses within the most accessible and well-known locations in the village (see Figure 6). Small public spaces and pocket parks with a scenic view of the village were also designed to create a sociable and healthy environment for the residents (see Figure 7). The physical interventions to improve semi-public and public realms helped to re-create a better living environment to organize everyday activities. The moms' community describes the space:

> This space has been useful especially for the elderly women . . . they use the space for daily purposes and for interacting with their friends. Even though it is a small space we are happy to get an additional square metre for ourselves.

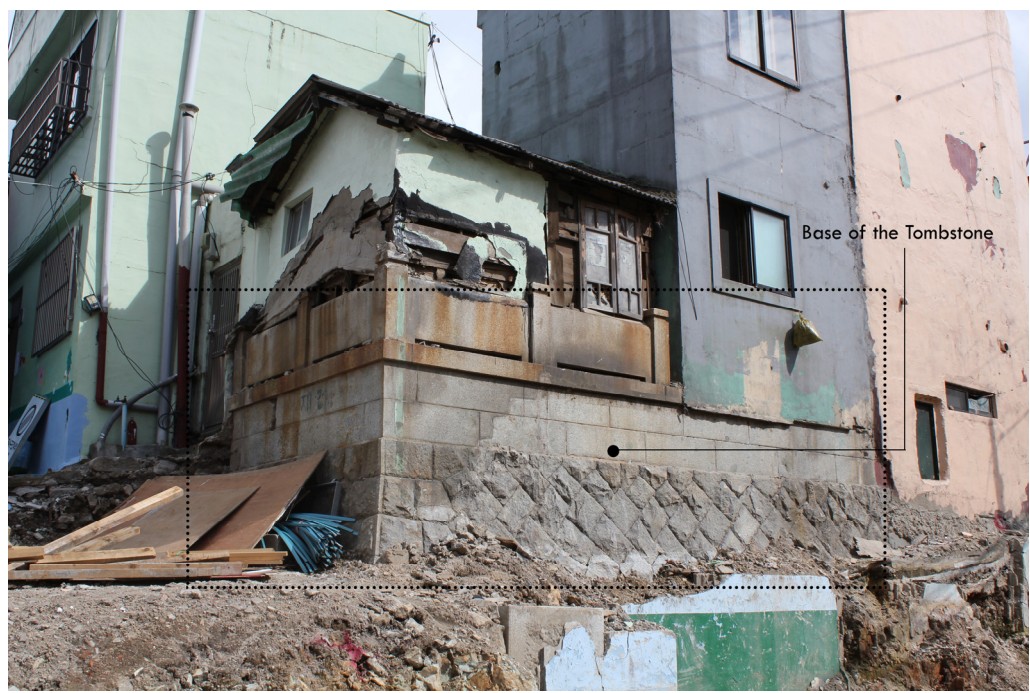

**Figure 5.** House constructed on a tombstone base.

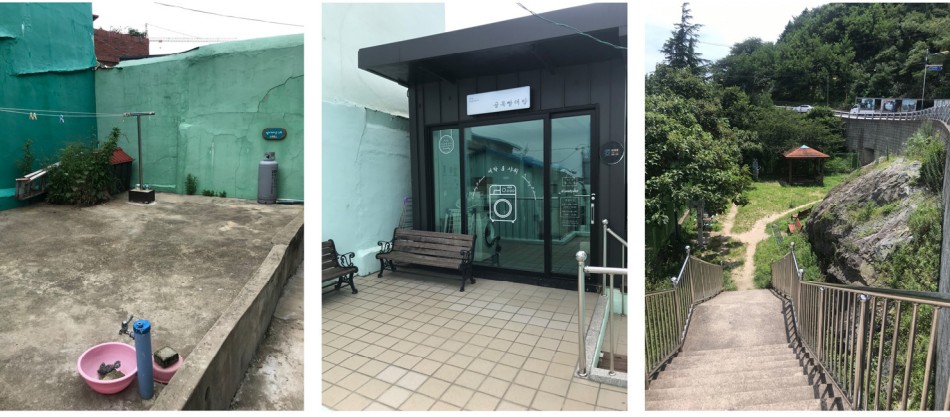

**Figure 6.** Semi-public urban pocket, laundry house and pocket park.

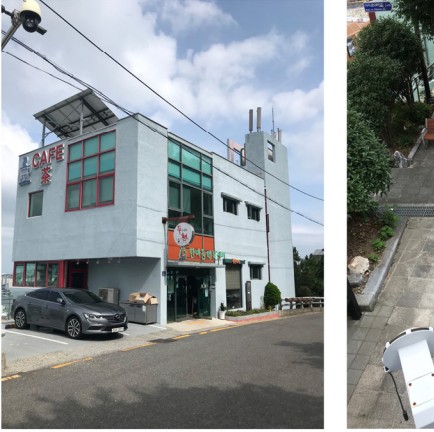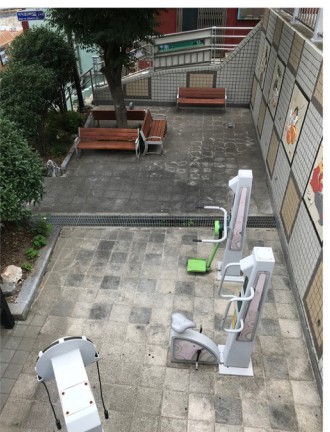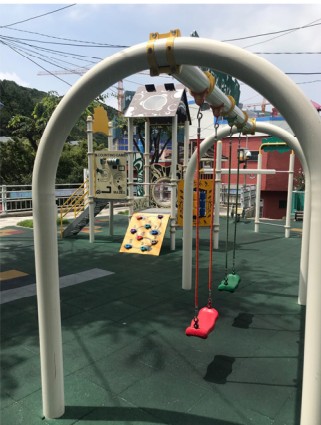

**Figure 7.** Happy Village Community Centre and small community spaces.

Safety and Crime Prevention

Owing to the steep, sloped terrain, most of the neighborhoods in the village are connected through long, narrow staircases. With the increase in the elderly population in recent years, the mobility across these steep passages has reduced on account of rising concerns of safety and security. Under the Crime Prevention through Environmental Design project, handrails, streetlights, safety mirrors and emergency bells were installed in all the inner routes for ease of movement and to create a sense of security in every neighborhood. The local police have established a small kiosk for surveillance to prevent any criminal activity due to a high concentration of vacant or abandoned houses. A new bus stop was also designed along the main street by demolishing a vacant house and includes heating benches to keep the residents warm during winters. A small plaza for holding community events was also constructed next to the bus stop (see Figure 8). One local government official emphasized the concerns of safety:

> The project made sure to increase surveillance and make the neighbourhoods brighter for safety purposes. Also, the demolition of some abandoned houses and strengthening of neighbourly relations . . . helped in creating a neighbourhood watch indirectly.

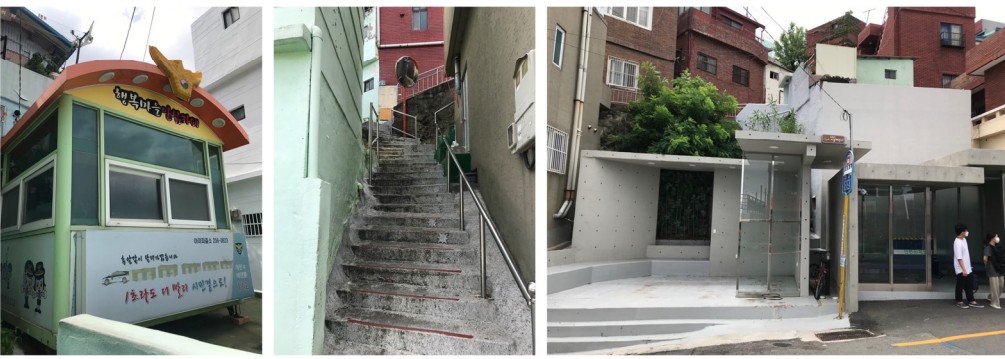

**Figure 8.** Police kiosk, street safety measures and village bus stop.

Conservation Strategy

As mentioned earlier, the history of Ami-dong has played an important role in shaping the lives of the residents and the spaces where they live. The regeneration initiative considered the preservation of remnants of historic elements in the village to be a priority as the tangible and intangible heritage reflects the identity of the residents. Integrating this objective within the regeneration plan, measures were taken to restrict the physical upgradation of the built environment based on passive strategies to preserve the existing spatial structure. A historical buffer zone, including the village and the surrounding area, was introduced under the village archiving initiative to restrict any damage to the tangible assets. The local government is working towards the recognition of tangible objects, artefacts and buildings associated with the Japanese Colonial period and the Korean War period and the intangible cultural heritage of the village.

3.3.3. Cognitive Responses—Physical Imageability

Place Theming

Over the years, residents have obliterated the Japanese cemetery by flipping the tombstones upside down, erasing the names on the graves and cementing the tombstone structure out of guilt. These efforts portray the intention to transform the cemetery into a living space. For changing this negative perception, a dark tourism initiative was introduced, and the village was named "Ami-dong Tombstone Village", thus replacing the past narrative with a sensitive interpretation that glorifies the lives of refugees and migrants. The moms' community conveys the importance of its identity with the dark history as an asset.

> Our village history brings back painful memories . . . but it is the truth, and we would like to share our story with everyone. The new image has given us a way to do so.

Landmarks, Cultural Facilities and Public Art

An historic tour trail connecting all the historic buildings and tourist facilities within the village was created to give the visitors a visual experience of the village's bittersweet history. The village entrance showcases photographs portraying life in the aftermath of the Korean War. The "Tombstone House"—an iconic wartime structure in the trail dates to the early 1950s and serves as an urban reminder of the life, culture and memory of war refugees in the village. Recently, vacant houses around the Tombstone House were remodeled as mini-museums that display artefacts and antiques collected from war refugees (see Figure 9). The trail also includes the Ami Cultural Learning Centre that houses a community café and viewing platform, a gallery displaying photographs and artifacts that belong to the famous documentary photographer Choi Min-Shik, who documented the lives of refugees in the downtown areas of Busan after the Korean War, a community center and an urban regeneration center (see Figure 9).

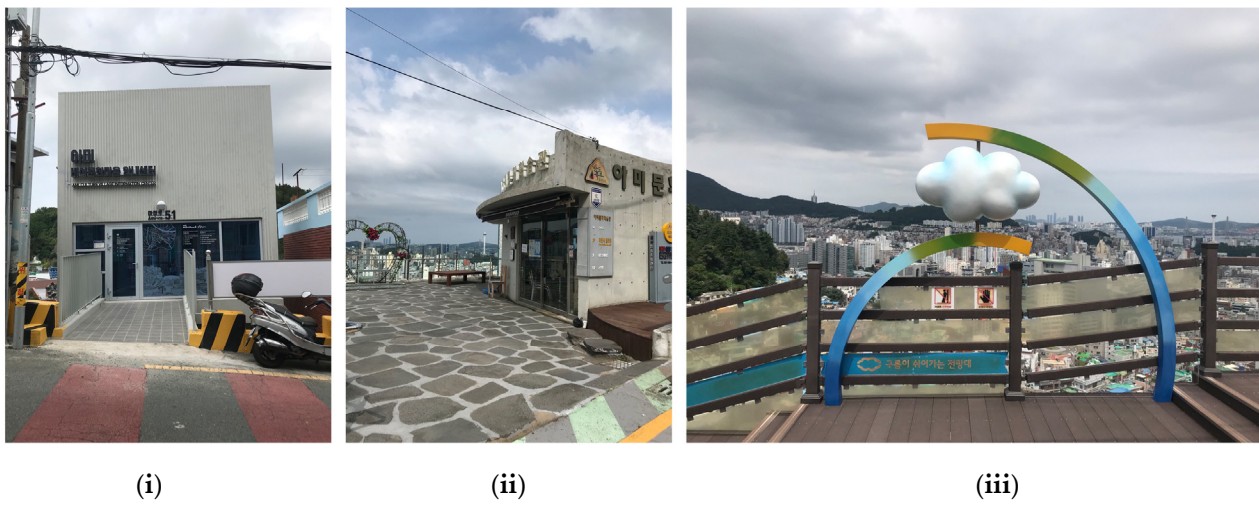

**(i)**                                   **(ii)**                                   **(iii)**

**Figure 9.** (**i**) Information Centre, (**ii**) Ami Cultural Learning Centre, (**iii**) Viewing Platform.

Painted murals on history and local identity were introduced as a part of the cultural regeneration efforts to enhance the vitality and vibrancy of the dull, congested alleyways. The trail also includes a village map at every tourist spot, resting areas and viewpoints with a scenic view of the entire village. This historic trail and the places re-created in the village during recent years help the tourists to easily find their way through the crooked alleyways and narrow streets. Visitors can find the remains of the tombstones scattered around the houses and alleyways (see Figures 10 and 11), stirring a bit of controversy as few of the residents have opposed the new image that overthrows their efforts of rooting out the agonizing past. Despite these claims, the new image is an authentic interpretation of the local identity, reminding of the individual and collective accomplishments of the residents. The new image has attracted at least 10% of the tourists visiting the neighboring Gamcheon Culture Village, which has gained global attention in recent years.

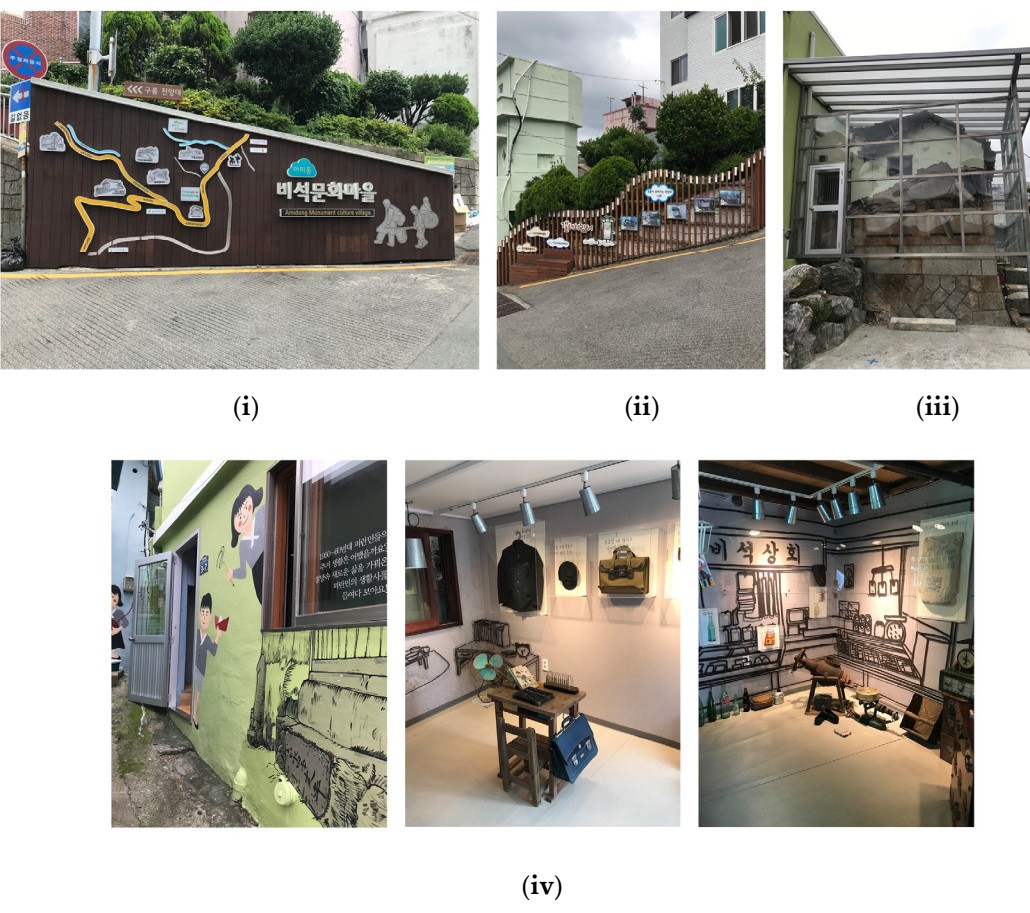

**Figure 10.** Historic Trail: (**i**) Village Entrance, (**ii**) Photo Trail, (**iii**) Tombstone House, (**iv**) mini-refugee museums.

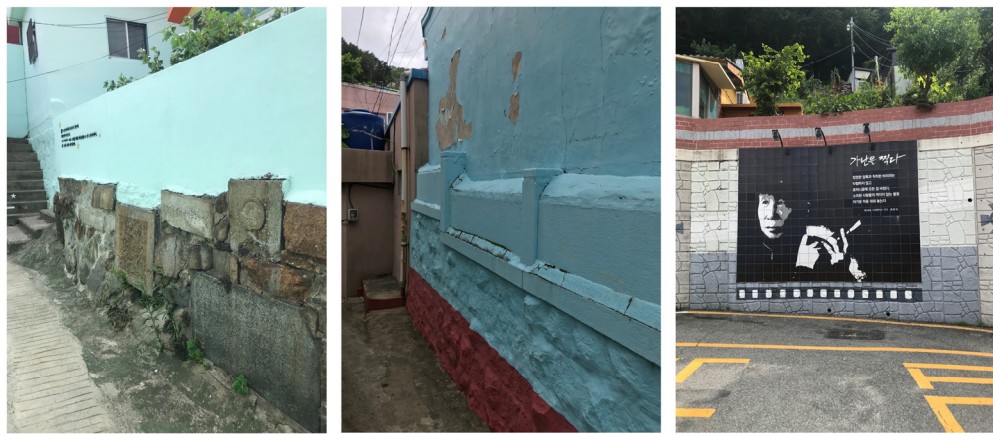

**Figure 11.** Public Art and remains of the tombstones along the alleyways.

Social Imageability—Festivals and Events

Since 2014, the Ami Moms community and the youth in the village have annually organized the "Let's Hang Out in Ami-dong" Festival to encourage resident interaction through various activities and to promote local businesses (see Figure 12). The Busan Christian Social Welfare Centre, Busan City Urban Regeneration Centre and the urban regeneration project have financially assisted and worked along with the residents to organize the festival every year. The festival began as a small event in 2010 and has become a village tradition ever since. The festival hosts a flea market, a historical tour, cultural

events and street cosplay to encourage people from nearby villages to visit Ami-dong. Even the village elderly have played an active role in this event by hosting various entertainment programs highly regarded by other residents and visitors.

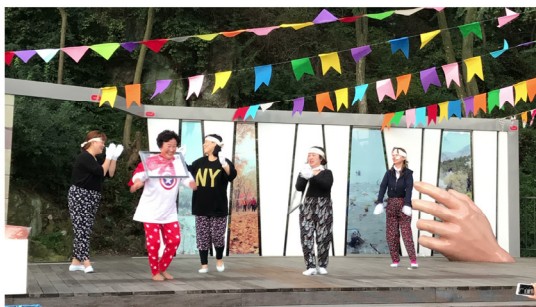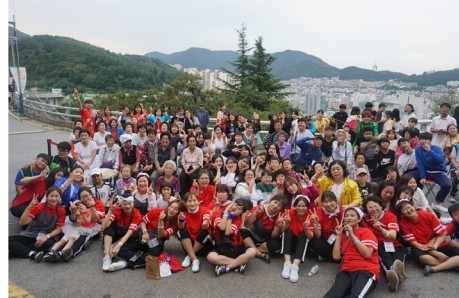

**Figure 12.** Let's Hangout in Ami-dong event (2014).

The village children developed a negative attitude and were reluctant to be identified with the village given its dark history. To change their perspective, the regeneration project helped the children participate in the village events and activities. Presently, the village children play an active role in organizing events and programs by supporting other residents (see Figure 13). The chief architect takes great pride in being able to help these children.

I personally feel proud of these children. They have changed a lot since the project helped them to do everything they wanted to. Now, they are actively being a part of every event in the village.

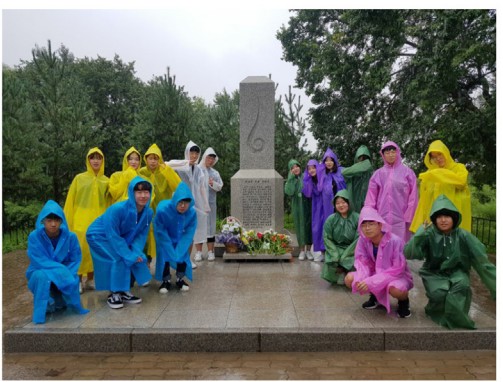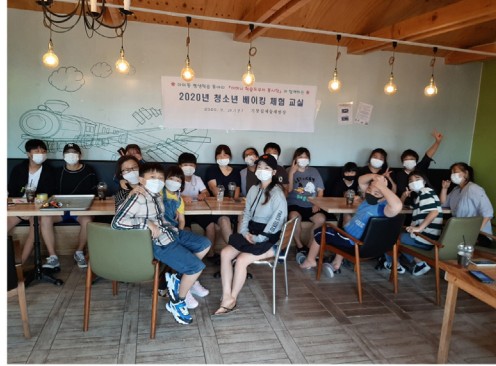

**Figure 13.** Children participating in a travel group program and other events.

The moms' community also reflects on the change in the children's attitude and expresses its gratitude.

We are happy to see this change in our children's perspective after the regeneration project. Our children are not embarrassed . . . but identify themselves as a part of the village. They participate and organize every event along with us.

### 4. Discussion

Sense of place and community dimensions within design and planning processes have gained attention over the recent decade with the growing concerns of social sustainability, especially in disrupted and changing urban landscapes. Prior studies emphasize these dimensions in urban changes (regeneration) by exploring concepts of tangential attachments [18], rescue geographies [49], local alliance [50], etc. Illustrating such contextualizations in practice, research related to case studies on place sensitivity and community involvement in urban regeneration, especially in Asian context, have increased significantly [37,50,51]. Such discussions emphasize the merits or consequences, i.e., the outcomes

of context-sensitive planning rather than the process. Also, such studies in connection with aging societies in declining contexts have been overlooked. To address these gaps, the present study illustrates the detailed process of integrating place sensitivity and sense of community within urban regeneration plans. Based on the case of Ami-dong Tombstone Village, a declining urban pocket with rising aging population in the downtown of Busan, the study explores physical elements, activity, meaning and place attachment integrated in the place change process (regeneration), using place-based vocabulary as suggested by Depriest-Hricko and Prytherch [17]. The article sought the three future directions suggested by Lewicka [16] to build a set of keywords within the "people–place–process framework of place attachment" (see Figure 1). Using the framework, the study provided insights on how sense of place has been articulated through physical, social, cultural and economic regeneration of Ami-dong (see Table 1).

**Table 1.** Integrating Sense of Place Elements within Regeneration Framework.

| Physical Regeneration | Social Regeneration |
|---|---|
| • Housing repair | Community involvement and participation |
| • Semi-public spaces | Revival of social network |
| • Pocket parks | Collective activities |
| • Children's play areas | Daily routines and interactions |
| • Community centers | Safety and comfort (CPTED) |
| • Community facilities (laundry house, bus stop, hair salon, etc.) | |
| • Tourist infrastructure (information center, viewing platforms, museums, etc.) | |
| **Cultural Regeneration** | **Economic Regeneration** |
| • Annual village festival | Small businesses and local entrepreneurship |
| • Small-scale programs and activities | Tourism promotion—Dark tourism strategy |
| • Conservation plan | |
| • Preservation of history—local heritage recognition | |
| • Public art | |

### 4.1. The Overall Process

The people-centered process adopted for the regeneration of Ami-dong has helped to formulate new strategies based on the issues and needs of the residents. By mobilizing different actors, including local government, district government, urban regeneration center officials and community members, comprehensive and integrated strategies for improving living conditions were formulated. The empirical findings of the study based on the regeneration proposals are as follows:

#### 4.1.1. Social Regeneration through Affective Response: Community Makes Everything Possible

First, although the definition of urban regeneration encompasses physical, social, economic and environment improvement, in most of the regeneration projects, physical and cultural renewal-based strategies are prioritized. The present study changes this approach by explaining the importance of involving community and local stakeholders in

the decision-making process. Despite the disrupted background, close ties were observed among the people as most of them have been long-term residents and have, in some cases, lived/worked together for more than 50 years. The regeneration project further strengthened these affective bonds between people and their village by involving them in the planning process. This interactive planning method has helped designers to identify the residents' needs, concerns and expectations as well as how they sense, feel or experience these places (see Figure 14).

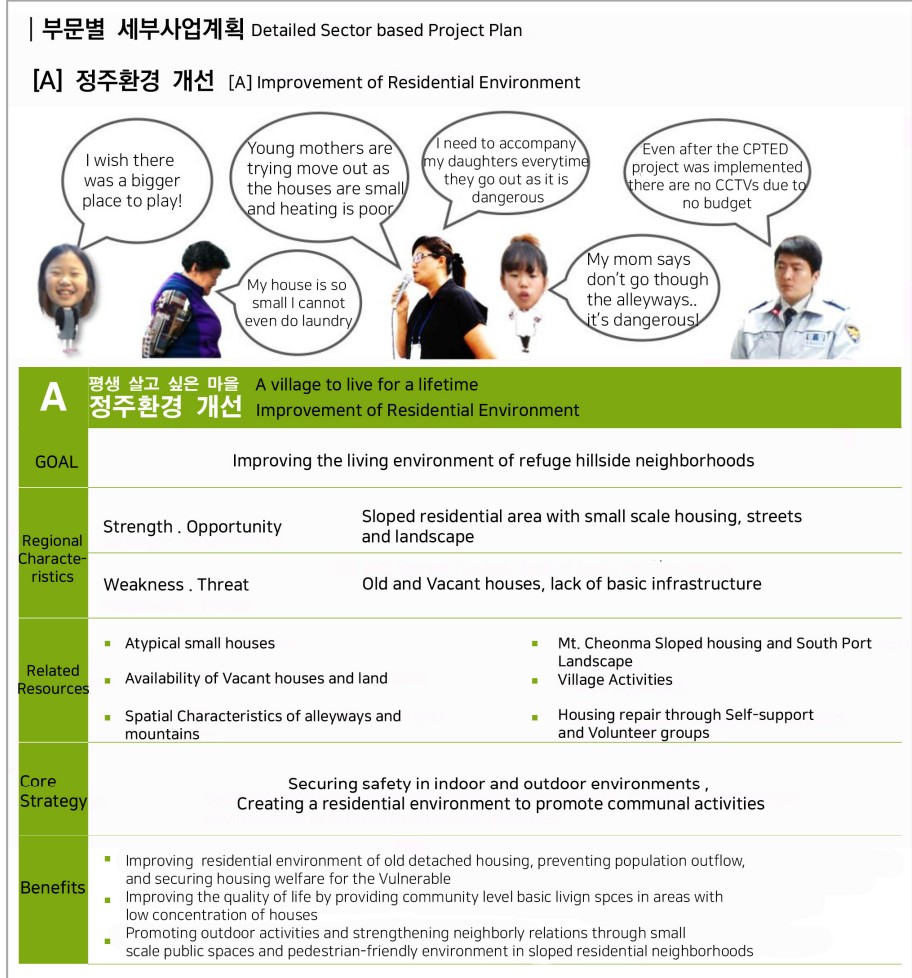

**Figure 14.** How people's dialogues were documented and used for making regeneration proposals [48]. Adapted with permission from Amichojang Dosijaesaeng Peurojekteu [Ami-Chojang Urban Regeneration Project]. 2016, by S.-K., Woo.

Even though the community's "voice and choice" [36] are subjected to change, they provide answers to "what kind of places are people seeking"(see Figure 14). Designing community facilities by drawing the "wants" and "don't wants" of the residents during the recent years has helped in creating places that have become a part of the residents' everyday settings. The routine communal activities and events in these spaces preserve the re-created social patterns and provoke a sense of togetherness, articulating dimensions of self and others as "I, my people, in our place". The Ami moms' community head explains:

> The project has brought us together . . . today we are not just a community but a family. We meet daily, enjoy community activities and celebrate every event . . . small or big . . . together irrespective of age.

### 4.1.2. Physical and Economic Regeneration through Behavioral Response: Spaces of Everyday Life and Personalization

Second, the urban regeneration initiatives have focused on improving and re-creating the lived spaces of the village by understanding the everyday life and space. As Inui Kumiko suggests, for finding ways to create desirable living spaces, we must focus on "what or why to create" rather than "how to create" by realizing "what kind of places are people looking for" [52]. From the invisible interactions and differences in uses observed during the surveys, proposals on improving the quality of little spaces within the neighborhoods were prioritized. Refining everyday landscapes and addressing the functional concerns of the place by establishing necessary infrastructure has unpacked ways for easily performing daily activities. Rapoport explains that behavioral settings are largely interpreted through people's personalization and willingness to participate in the process of improving their environments [39]. The semi-public urban pockets or small public spaces have helped the residents to impose their character and personality in the place. It is not the magnitude of these spaces but the significance of human scale, which fits in the physical setting and is filled with significances of the people using them. The moms' community believe these spaces provide an additional square footage and a new living room for the village elderly.

> Even though outsiders might not view these spaces as a huge upgrade we believe these spaces have given us additional room for our residents. Also the village elderly frequently interact and look after each other through these tiny spaces.

The overall physical regeneration proposals were based on improving the existing built environment through restoration and repair rather than creating new structures that might create a sense of strangeness. New communal facilities were created by adaptive re-use of vacant/abandoned houses. Community centers built during recent years have been established as nodal points within the existing physical fabric for connecting the detached neighborhood structure.

### 4.1.3. Cultural Regeneration through Cognitive Response: Dark History Becomes a Public Asset

Third, in some of the proposals, for instance, the historic place theming strategy, designers and users differently interpreted and reacted to the physical environment. While design and urban professionals considered the historic significance as a mnemonic for communicating the values, aspirations, memories and experiences of the community, few residents considered it to be an impediment. The complexity of these perceptions was resolved using an empathetic depiction of the village's dark history by replacing the images of death and war with glorious narratives on struggles and achievements of refugees and migrants. Rather than replacing the identity of the village with an insignificant theme, the regeneration project created a place-sensitive identity that encompasses the local community's experiences and memories, narrating the stories of the past through a brand-new urban lens, thus illustrating a distinction between Ami-dong and its surroundings while preserving continuity of its history in the present context. This new image helped to change the negative perceptions of the residents and has become a channel for narrating their stories.

> These changes have helped us to create a strong identity ... we were always forgotten or hidden (prior) ... even though we have a dark and sad history ... we are proud to be residents of Ami-dong and this (new image) is the way to narrate our stories.

## 5. Conclusions

In light of the ongoing challenges of urban decline and shrinking population in Korea, policymakers and designers are urging to find new ways to implement regeneration plans based on community planning approaches. Community engagement in the decision-making process can help in improving environments based on visions of people and can revive a "sense of life" in an otherwise declining background. This approach can help in

formulating context-specific solutions rather than rational strategies that ignore people's perceptions. Therefore, this study attempts to highlight the case example of Ami-dong Tombstone village in Busan, South Korea to explain the process of articulating the residents' "sense of place and belonging" within regeneration plans, using the "people–place–process framework of place attachment" reframed with key concepts and definitions provided in previous research and future directions suggested by Lewicka [29]. Using the three-dimensional framework, the study evaluates the role of social capital, and community empowerment in strengthening place associations and motivating people to participate in changing their environments. The processes of place-making and place-theming, based on the residents' perceptions, interactions, practices and symbols, enables a sense of distinctiveness and continuity in local place identity irrespective of positive/negative conceptions.

The results of the study show that the regenerations plans implemented over the years have helped to create a sense of identity, safety and security, comfort and well-being by creating or improving everyday physical settings in public, semi-public and private realms of the village. The overall regeneration initiatives appear to be minor but convey how people's way of making sense of places can help in creating meaningful environments. These processes signify the preservation of people, their community and the place that they proudly call "home", even against a disrupted background [19]. The case study is limited to qualitative explorations of place sensitivity aspects in urban regeneration but clearly documents the process of creating place-sensitive urban plans. Such practical commentaries on local sense of place and community participation can help urban professionals and architects with the successful planning of places, especially within the context of disruptions and incivilities. Future studies can focus on comparative analysis of place-sensitive planning and rational planning through resident satisfaction and opinion surveys.

**Author Contributions:** Conceptualization, methodology, formal analysis, investigation, writing—original draft preparation, S.K.; writing—review, supervision, I.-H.L. Both authors have read and agreed to the published version of the manuscript. All authors have read and agreed to the published version of the manuscript.

**Funding:** This work was supported by BK21 FOUR Program by Pusan National University Research Grant, 2021.

**Informed Consent Statement:** Informed consent was obtained from all subjects involved in the study. Written informed consent has been obtained from the subject(s) to publish this paper.

**Data Availability Statement:** Publicly available demographic datasets were used in this study. This data can be found here: [https://kosis.kr/eng/].

**Acknowledgments:** The authors would like to thank Woo Shin-Koo for providing relevant data, insights and expertise that greatly assisted this research. The authors are also thankful to the Ami-Moms community for their cooperation and time during the interviews.

**Conflicts of Interest:** The authors declare no conflict of interest.

## Notes

1. The Korean word "Maeul" translates to "village" in English and is used in an urban context to designate neighborhoods in downtown areas.
2. "Wondosim" or the original city center of Busan, corresponds to the first settlement area of the city near the Busan Port. The origins of the city began during the Joseon dynasty and was later occupied by the Japanese in 1910. After the Korean War, this area served as a major commercial and industrial center for the city. In the 1990s, the development of the city shifted towards the east, and "Seomyeon" and "Haeundae" became the new commercial districts replacing the original city center. Presently, this area is known as the "old downtown of Busan".
3. The tombstones of the deceased were erected on a large base under which the ashes were stored. After the Korean War, refugees obliterated the headstone and used the base as a plinth for constructing houses.

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
