# Peer review of "Rethinking Sense of Place Interpretations in Declining Neighborhoods: The Case of Ami-dong Tombstone Cultural Village, Busan, South Korea"

_societies, doi:10.3390/soc13020030_

Round 1

Reviewer 1 Report

Dear authors,

The research presented is an illustration of the strongest and, thus, critical role of collective action. In the case presented this action results in a new urban dynamic and solidarity. In order to improve the paper my advice is to you to:

1. Clarify the originality fo the research within urban studies and, therefore, other cases of urban regeneration through collective action/social capital. 

2. Clarify figure 1. This schema is not clear in its structure and information. For instance if one of the dimensions is social, why is it in the personal dimension? The meaning of 'process' dimension is not clear. In fact the utility of this schema is not clear.

3. The paper is rather descriptive than analytical. It is true that the conclusion allows the interaction between theory/concepts and the empirical set. But, still, the authors should try to be more analytical during the paper. The way as it is now, looks more as a report.

Author Response

Dear reviewer,

Thank you for giving us the opportunity to submit a revised draft of our manuscript ‘Rethinking sense of place interpretations in planning declining neighborhoods: a case of Ami-dong Tombstone Village, Busan, South Korea’. We appreciate your insights and feedback on our research article. The comments given in the first review have helped us to make suitable revisions to our study. We hope the following responses will cater to the inquiries and suggestions received from the review.

Thank you.

Response to reviewer's comments:

Please refer to the attached manuscript file to view the changes made in the article based on the comments.

Comment 1:  Clarify the originality fo the research within urban studies and, therefore, other cases of urban regeneration through collective action/social capital. 

Response: Page 3 (103-118)  - as suggested by the reviewer we have tried to include few successful cases related to urban regeneration and environmental improvement. Additional references 37,38 were added in this section.

"One study based on community-led regeneration in China reveals that strong social ties and communal identity in a deteriorating fishing community helped in revitalizing the physical fabric through participatory planning and design [37]. In Korea, collaboration of artists, residents and local government in culture-led regeneration programs helped in building neighbourhood trust, revitalizing the physical fabric, and strengthening people-place relationships that encouraged the residents to actively participate in future im-provement programs [38]. Another study related to the disorientation and reorientation among people after the F3 Tornado in Goderich, Ontario shows that the people-people and people-place relations were strengthened after the disaster through active and collective engagement of the community in the recovery and restoration process [22]. Studies on involuntary relocations due to conflict or development and rational planning for place improvements have shown that strong place associations have helped in re-establishing a sense of local identity despite the place transformations resulting from urban regeneration [18,21,23]. Therefore, irrespective of the contextual setting (ordered or disrupted), the relationship between people and places can be analysed by examining the organisation of space, meaning, time and communication within the perceived environment [39]."

----------------------------------------------------------------------------------------

Comment 2: Clarify figure 1. This schema is not clear in its structure and information. For instance if one of the dimensions is social, why is it in the personal dimension? The meaning of 'process' dimension is not clear. In fact the utility of this schema is not clear.

Response: 

  • As mentioned on page 3 (135-143), the suggestions for future studies recommended by Lewicka on people, place and process dimensions were considered. These suggestions were explored in this research using the people-place-process framework.
  • The relationship between people and places can be examined through the process dimensions. Also, these three dimensions work simultaneous as they are interconnected and cannot be separately analyzed. Therefore, the process dimensions are used as a main catalyst for understanding the dynamics of people-place relationships. This was showed through the figure 1 for understanding the complex nature of relationship between people-place-process dimensions through arrows in the framework.
  • The 'social' aspects are based on individual and collective responses of people towards a place. Thus, in the framework we tried to show the interconnections through arrows for easier understanding.

----------------------------------------------------------------------------------------

Comment 3: The paper is rather descriptive than analytical. It is true that the conclusion allows the interaction between theory/concepts and the empirical set. But, still, the authors should try to be more analytical during the paper. The way as it is now, looks more as a report.

Response: The paper uses a qualitative analysis method for understanding the process of integration of place-sensitivity in urban regeneration through participatory approaches. We tried to analyze the keywords within the framework in Figure 1 through dialogue and narration for understanding the benefits of place-sensitive approaches at the ground-level. The paper might appear to be a report but is a true interpretation and detailed analysis of the case study through primary and secondary data sources.

Reviewer 2 Report

Dear Authors,

Thank you for sending this paper to the journal Societies. The subject of the study is interesting, but there is a significant gap in this version. I wonder encouraged your guys to improve this manuscript. This reviewer has some errors regarding the clarity of the paper:

Comment 1: You mentioned place-related, physical setting, psychological place etc. Perhaps literature review is not enough. When you focus on planning and design, your refs. should consider natural factors and public health.  1)An ecological study of physical environmental risk factors for elderly falls in an urban setting of hong kong. Science of The Total Environment, 407(24), 6157-6165.  2)Modeling the impact of soundscape drivers on perceived birdsongs in urban forests. Journal of Cleaner Production, 2021, 292, 125315; 3) Attachment and identity as related to a place and its perceived climate. Journal of Environmental Psychology, 25( 2), 207-218.

 Comment 2: Line 76-78 This description is not suitable in an academic production. The same problem at Line 414-415

Comment 3: Person dimension. Perhaps more factor, not only personal value.

Comment 4: Figure 2-  recently years is not detailed description. I wonder know when these places created?

Comment 5: What sources of your history information? Whether including unofficial history or privately compiled history?

Comment 6: Figure 14 should be translated into English.

Comment 7: Limitation should be mentioned in Discussion section.

Comment 8: The Structure of Conclusion section should be improved.

Best regards

Author Response

Response to Reviewer 2 comments:

Dear reviewer,

Thank you for giving us the opportunity to submit a revised draft of our manuscript ‘Rethinking sense of place interpretations in planning declining neighborhoods: a case of Ami-dong Tombstone Village, Busan, South Korea’. We appreciate your insights and feedback on our research article. The comments given in the first review have helped us to make suitable revisions to our study. We hope the following responses will cater to the inquiries and suggestions received from the review.

Thank you.

Response to reviewer's comments:

Comment 1: You mentioned “place-related”, “physical setting”, “psychological place” etc. Perhaps literature review is not enough. When you focus on planning and design, your refs. should consider natural factors and public health.  1)An ecological study of physical environmental risk factors for elderly falls in an urban setting of hong kong. Science of The Total Environment, 407(24), 6157-6165.  2)Modeling the impact of soundscape drivers on perceived birdsongs in urban forests. Journal of Cleaner Production, 2021, 292, 125315; 3) Attachment and identity as related to a place and its perceived climate. Journal of Environmental Psychology, 25( 2), 207-218.

Response:

We have considered major references from the fields of environmental psychology, social psychology, sociology, urban studies and urban design for our research.

For our research we have considered many studies that include a multi-dimensional research understanding (For example: Carmona, M. (2018). Place value: place quality and its impact on health, social, economic and environmental outcomes. Journal of Urban Design, 24(1), 1-48; doi: 10.1080/13574809.2018.1472523)

Also, some of the references were taken from psychiatry (Fullilove, M. (1996). Psychiatric Implications of Displacement: Contributions From the Psychology of Place. The American Journal of Psychiatry, 153(12), 1516-1523; doi: 10.1176/ajp.153.12.1516)

While we appreciate the reviewer’s feedback, we think that including other research as suggested in this comment, the article may deviate the main focus of the study. 

--------------------------------------------------------------------------------------------------------------------------------------------

Comment 2: Line 76-78 This description is not suitable in an academic production. The same problem at Line 414-415

Response: The lines have been corrected as suggested

Line 76-78:

Original: It is a dynamic human-environment relationship involving emotions, feelings and experiences within a place built through unselfconscious and self-conscious processes [31]. 

Revised: It is a dynamic human-environment relationship involving emotions, feelings and experiences within a place [31].

Line 414-415:

Original: To change their perspective, the regeneration project helped the children form a small group to organize and participate in the village events and activities. These interventions helped in changing the children’s negative perceptions. 

Revised: To change their perspective, the regeneration project helped the children participate in the village events and activities.

--------------------------------------------------------------------------------------------------------------------------------------------

Comment 3: Person dimension. Perhaps more factor, not only ‘personal value’.

Response: We have included additional factors relevant to our study and used during the analysis under the person dimension

Comment 4: Figure 2-  “recently years” is not detailed description. I wonder know when these places created?

Response: Revised the figure caption for clarity

Figure 2: Map of Busan showing the Original City Centre (Left), and Location of Ami-dong and places created/recreated in recent years between 2013 and 2020 (Right)

 -------------------------------------------------------------------------------------------------------------------------------------------

Comment 5: What sources of your history information? Whether including unofficial history or privately compiled history?

Response:

Primary Data -

  • Through our site investigations, we explored the history of the village through the information provided in the historic photo trail, the landmark buildings, museums, photo gallery and information center in the village.
  • We also conducted interviews of the chief architect involved in the urban regeneration of the village who is also working on documenting the history of the village. Through the interview we could find relevant information and other secondary sources.

Secondary Research -

  • The history of the village was researched through reference 45. This book gives a detailed account of historic development of the village from 1950s to present. The book also includes personal accounts and place perceptions of the people residing in the village since 1950s.
  • Also, this section uses reference 46 which documents of all the planning policies and projects implemented in the village since 1960s to present.

Comment 6: Figure 14 should be translated into English

Response: As suggested we have translated Figure 14 to English

Comment 7: Limitation should be mentioned in Discussion section.

Response: Limitations and future suggestions were also included to the manuscript as suggested on Page 15

“The case study is limited to qualitative explorations of place sensitivity aspects in urban regeneration but clearly documents the process of creating place-sensitive urban plans. Such practical commentaries on local sense of place and community participation can help urban professionals and architects for successful planning of places especially within the context of disruptions and incivilities. Future studies can focus on comparative analysis of place-sensitive planning and rational planning through resident satisfaction and opinion surveys.”

Comment 8: The Structure of Conclusion section should be improved.

Response: We tried to restructure the conclusion section as suggested by the reviewer.

Revised Conclusion:

In light of the ongoing challenges of urban decline and shrinking population in Korea, policy makers and designers are urging to find new ways to implement regeneration plans based on community planning approaches. Community engagement in the decision-making process can help in improving environments based on visions of people and revive a ‘sense of life’ in an otherwise declining background.  This approach can help in formulating context-specific solutions rather than rational strategies that ignore people’s perceptions. Therefore, this study attempts to highlight the case example of Ami-dong Tombstone village in Busan, South Korea to explain the process of articulating the residents’ ‘sense of place and belonging’ within regeneration plans using the ‘people-place-process framework of place attachment’ reframed with key concepts and definitions provided in previous research and future directions suggested by Lewicka [29]. Using the three-dimensional framework, the study evaluates the role of social capital and community empowerment can help in strengthening place associations and motivating people to participate in changing their environments. The processes of place-making and place theming based on the residents’ perceptions, interactions, practices, and symbols enable a sense of distinctiveness and continuity in local place identity irrespective of positive/negative conceptions.

The results of the study show that the regenerations plans implemented over the years have helped in creating a sense of identity, safety and security, comfort, and well-being by creating or improving everyday physical settings in public, semi-public, and private realms of the village. The overall regeneration initiatives appear to be minor but convey how people’s way of making sense of places can help in creating meaningful environments. These processes signify the preservation of people, their community, and the place that they proudly call ‘home’ even against a disrupted background [19].  The case study is limited to qualitative explorations of place sensitivity aspects in urban regeneration but clearly documents the process of creating place-sensitive urban plans. Such practical commentaries on local sense of place and community participation can help urban professionals and architects for successful planning of places especially within the context of disruptions and incivilities. Future studies can focus on comparative analysis of place-sensitive planning and rational planning through resident satisfaction and opinion surveys.

Reviewer 3 Report

I found this article to be well written, easy to follow, and highly relevant, although I thought it was a little light in the discussion.

I found the introduction to set the scene well in terms of a) explaining place, b) the framework to be discussed and c) a few recent examples that have attempted to communicate place. While there is no literature review evident, it doesn't feel necessary. There is a wide range of relevant literature discussed throughout, and instead a thorough explanation of the case study and it's relation to method/theory and other literature are evident.
The case study information is informative and thorough.

It could be the structure of the article, but I found the discussion to rather quickly summarise the very in-depth case study. The discussion points raised are interesting however, and I don't think there are any major changes needed here. Perhaps, you could add in an interview quote or two that exemplifies the discussion point? This might just tie together the people, sense of place framework and the case study nicely.

Overall, I found it intriguing to learn more about Ami-dong, as I had recently travelled to Gamcheon Cutural Village just last year. The article does a good job of providing the historical context, the regeneration process and it's relation to the framework. I think there is an interesting and relevant contribution made here.

Author Response

Dear reviewer,

Thank you for your providing your valuable feedback for our manuscript. We are grateful for your insightful comments regarding every section of the manuscript. We have incorporated some changes to reflect on the suggestions provided to us.

Response to Reviewer's comments:

Comment: Perhaps, you could add in an interview quote or two that exemplifies the discussion point? This might just tie together the people, sense of place framework and the case study nicely.

Response: In the discussion section we have included some quotes from the interviews under each subhead in our revised manuscript.

Round 2

Reviewer 1 Report

Dear authors,

the paper still have some problems regarding conceptual, and empirical findings, maintaining its essencial descriptive and report approach mode.

Author Response

Dear reviewer,

Thank you for your providing your valuable feedback for our manuscript. We are grateful for your insightful comments regarding every section of the manuscript. We have incorporated some changes to reflect on the suggestions provided to us.

Response to Reviewer's comments:

Comment: the paper still have some problems regarding conceptual, and empirical findings, maintaining its essencial descriptive and report approach mode.

Response: Thank for raising this point. This research has been based on people's interpretation of sense of place (place- Ami-dong) through dialogues and daily scenes documented during surveys and interviews. This qualitative analysis method has helped us to interpret the on ground scenario of the case selected and has a descriptive section within the case study.  

Reviewer 2 Report

Very unique research done.

Most of my comments have been clarified in this Revision.

I recommend this publication after one minor thing:

Ref. Fullilove, M. (1996). is too old. Please update a new one.

Author Response

Dear reviewer,

Thank you for your providing your valuable feedback for our manuscript. We are grateful for your insightful comments regarding every section of the manuscript. We have incorporated the changes in our new manuscript to reflect on the suggestions provided to us.

Response to Reviewer's comments:

Comment: Very unique research done. Most of my comments have been clarified in this Revision. I recommend this publication after one minor thing: Ref. Fullilove, M. (1996). is too old. Please update a new one.

Response: Thank for raising this point. Along with Fullilove (1996) an updated reference has been added to the revised manuscript.

"Seamon, D., & Sowers, J. (2008). Place and Placelessness, Edward Relph. In P. Hubbard, R. Kitchin, & G. Valentine (Eds.), Key Texts in Human Geography (pp. 43-51). London: Sage."